# Imaging Mass Spectrometry: A New Tool to Assess Molecular Underpinnings of Neurodegeneration

**DOI:** 10.3390/metabo9070135

**Published:** 2019-07-10

**Authors:** Kevin Chen, Dodge Baluya, Mehmet Tosun, Feng Li, Mirjana Maletic-Savatic

**Affiliations:** 1Department of Biosciences, Rice University, Houston, TX 77030, USA; 2Department of Pediatrics, Baylor College of Medicine, Houston, TX 77030, USA; 3Jan and Dan Duncan Neurological Research Institute at Texas Children’s Hospital, Houston, TX 77030, USA; 4Chemical Imaging Research Core at MD Anderson Cancer Center, University of Texas, Houston, TX 77030, USA; 5Center for Drug Discovery and Department of Molecular and Cellular Biology, Baylor College of Medicine, Houston, TX 77030, USA; 6Department of Neuroscience and Program in Developmental Biology, Baylor College of Medicine, Houston, TX 77030, USA

**Keywords:** neurodegeneration, metabolomics, biomarkers, imaging mass spectrometry

## Abstract

Neurodegenerative diseases are prevalent and devastating. While extensive research has been done over the past decades, we are still far from comprehensively understanding what causes neurodegeneration and how we can prevent it or reverse it. Recently, systems biology approaches have led to a holistic examination of the interactions between genome, metabolome, and the environment, in order to shed new light on neurodegenerative pathogenesis. One of the new technologies that has emerged to facilitate such studies is imaging mass spectrometry (IMS). With its ability to map a wide range of small molecules with high spatial resolution, coupled with the ability to quantify them at once, without the need for a priori labeling, IMS has taken center stage in current research efforts in elucidating the role of the metabolome in driving neurodegeneration. IMS has already proven to be effective in investigating the lipidome and the proteome of various neurodegenerative diseases, such as Alzheimer’s, Parkinson’s, Huntington’s, multiple sclerosis, and amyotrophic lateral sclerosis. Here, we review the IMS platform for capturing biological snapshots of the metabolic state to shed more light on the molecular mechanisms of the diseased brain.

## 1. Introduction

Imaging mass spectrometry (IMS) has emerged as a powerful molecular imaging technology, enabling us to map, with high molecular specificity and sensitivity, the spatial distribution of small molecules in tissues. As such, has allowed us to accelerate scientific discoveries on the role small molecules and metabolites play in health and disease. These small molecules, such as lipids, sugars, neurotransmitters, amino acids, and xenobiotics, are not readily detected by traditional methods of molecular imaging, such as microscopy or in situ hybridization. IMS allows us to visualize, map, and analyze hundreds of these molecules at the same time, in a single, label-free sample [1,2,3,4,5]. Thus, it differs from conventional mass spectrometry, which has revolutionized the worlds of omics sciences and drug metabolism, disposition, and development, but has been mostly used for studies of analytes that have been extracted from fluids or tissues [6], while lacking the spatial information of the metabolite distribution. IMS also allows correlation between the abundance and localization of specific compounds in tissue samples with the histological images obtained from the same or adjacent tissue sections [7]. Such knowledge, in turn, is important in elucidating the function of small molecules within complex biochemical pathways and their roles in health and disease [8]. 

Importantly, in contrast to the simplification of complex biochemical pathways in reductionist viewpoints, IMS allows a systems approach that can be very valuable to integrating the spatial component of small molecules into our understanding of their role in cellular and intercellular connections [6,9]. Furthermore, hundreds of metabolites can be analyzed and mapped at one time, without the need for labels, staining, and radioactive trackers, to distinguish the different metabolites of interest [1]. Thus, IMS brings new dimensions to molecular imaging and places it at the forefront of many applications in metabolism and small molecule/drug discovery research [7,10,11,12,13]. This review focuses on the existing IMS technologies and presents some of their potential applications in neurodegenerative diseases. 

## 2. Imaging Mass Spectrometry: Advantages and Disadvantages

For a longwhile, the brain has been considered a homogeneous structure in terms of metabolite distribution. After all, all cells depend on energy metabolism to survive. Without accounting for the regional specificity of metabolites [14], their identification and quantification in a given sample has failed to detect spatially distinct metabolic alterations that may have been important for understanding the disease pathology. The recognition of these shortcomings has increasingly led to harnessing the imaging and profiling capabilities of IMS in the field of neuroscience, in order to better understand and profile the metabolic changes in neurodegenerative diseases, as well as to find potential biomarkers for their diagnosis and monitoring [15]. 

Interestingly, over the past few years, the lipidome has gained much traction in research on neurodegenerative disorders, due to the importance of lipid signaling in various cellular pathways [16]. Irregularities of the lipidome, such as erroneous lipid metabolism and signaling, have not only been tied to diseases that involve large-scale metabolic dysfunction (e.g., diabetes, hypertension, atherosclerosis, diabetes), but also neurodegenerative diseases such as Alzheimer’s, Parkinson’s, and Huntington’s diseases, amyotrophic lateral sclerosis, and multiple sclerosis [17]. The nervous system is home to the most heterogeneous and distinct lipid classes in the entire body [18]. While lipidomics of the brain tissue has contributed much to our knowledge of the lipid content of the brain, IMS has brought forward the possible relevance of certain classes of lipids with certain brain functions. For example, the selective localization by IMS of C20 gangliosides in the molecular layer of the dentate gyrus, a region of the brain central to learning and memory [19], has pointed to a possible role of these lipids as key constituents of neurons, contributing to learning and memory. The relevance of this discovery is still to be elucidated; regardless, new investigations of specific roles of different classes of lipids are now possible with IMS technology. 

### 2.1. Advantages

The IMS has gained its prominence because of its ability to detect, relatively quantify, and map small molecules (<2 kDa) and the metabolites within a sample in situ, maintaining high spatial resolution and molecular specificity without the need for chemical labels, staining procedures, and molecular probes. IMS is one of the technologies producing big data and serves as an addition to the histologist’s toolbox: a complement to it, not a replacement for it. By integrating microscopy with IMS, the applications become almost limitless, and could be used to answer a variety of biologically and medically relevant questions. The spatial resolution of IMS-generated images approaches the cellular level, which is advantageous for studies of tissues composed of a variety of cell types, or of genetic chimeras, as the metabolic signals of neighboring cells can be different. Thus, a more comprehensive and holistic understanding of the genotypic, phenotypic, and metabolic responses of the tissue to disease pathology or to changes in the environment can be obtained with IMS in efforts to study complex disease biology. 

In addition, the distribution of xenobiotics, including drug species, can also be studied with IMS. The effects of exogenous species on metabolism and endogenous metabolites can accelerate finding the biomarkers and molecular links of a given disease [20]. Pharmaceutical research has typically used liquid chromatography in tandem with mass spectrometry (LC-MS) to conduct pharmacokinetic studies in preclinical trials on animal models, but the rise of IMS platforms promises to shorten the preclinical research flow. Traditional LC-MS fails to provide any information on spatial localization of a given drug, due to the excision and homogenization of tissue samples. In contrast, IMS is the most thorough and unbiased way to map the penetration of compounds of interest into tissues and their distribution throughout the body. This is most valuable for establishing pharmacokinetic properties, including accumulation in non-target tissues and excretion routes. Furthermore, IMS can also detect the bio-transformed metabolites of a drug for the unbiased determination of true, biologically active drug compounds and their toxic effects. Overall, IMS is a powerful, yet cost-effective technology that will enable distribution studies to be performed earlier in the drug discovery process, without any requirement for radiolabeled standards [20,21]. 

### 2.2. Disadvantages

Although a very valuable tool for metabolite detection, IMS has some caveats, many of which stem from the process of sample preparation [20]. While simultaneous detection of carbohydrates, proteins, lipids, nucleotides, matrix ions, and salt is one of the crowning achievements of IMS, this process also leaves this platform open to ion suppression, which occurs when the chemically-distinct natures of each metabolite impair the overall detection [22]. Namely, each metabolite impairs the desorption and ionization efficiency of every other molecule; thus, the aggregate effect results in more abundant metabolites being selectively ionized over less abundant ones, leading to the depletion of signals from low-abundance metabolites that may be of interest. One possible approach to overcome this pitfall is to remove a particular metabolite class; however, this can affect sample integrity, causing molecular diffusion and adduct formation, which ultimately leads to increased complexity of data [20,22]. Another disadvantage of IMS is sample degradation, which can occur as the tissue is thawed during sample preparation. Certain analytes are not stable at room temperature and may completely degrade, leading to misinterpretation of the data [22]. For matrix-assisted laser desorption ionization (MALDI) IMS, matrix application must be uniform across the entire sample to minimize image artifacts, and this may be difficult to achieve [20,22]. Furthermore, until sample preparation is standardized, with standard procedures for each class of metabolites, IMS data may not be fullyreproducible, due to batch effects. Alternative ionization methods have already proven effective in circumventing some of these problems. For example, secondary ion MS minimizes problems with matrix inconsistencies and analyte diffusion by not requiring matrix application at all [20,22]. Finally, with respect to data processing, IMS requires substantial computing capabilities, as the acquired data can reach upwards of several gigabytes, and corresponding processing times of several hours [20]. In sum, as sample preparation becomes increasingly standardized and ionization and desorption techniques continue to improve, IMS will continue to see a growth in popularity.

## 3. Ionization Methods

Many steps need to be conducted correctly to ensure optimal IMS results: sample preparation, sample desorption and ionization, mass spectrum analysis, and image production [20]. Depending on the biological class of the compound of interest, the ionization technique can be modified to best suit data acquisition [20,22]. The three ionization techniques most central to the existing IMS platform are MALDI, desorption electrospray ionization (DESI), and secondary ion mass spectrometry (SIMS) [22] (Table 1).

Either positive or negative ionization modes can be employed to trigger and detect particular ion formations. In positive ion mode, the molecules gain protons to become cations, whereas in negative ion mode, the molecules are deprotonated to form anions [20]. The three ionization techniques influence the lateral resolution in IMS, defined as the minimum pixel size needed to produce a detectable signal. For MALDI, lateral resolution is affected by the laser spot size, while for DESI, by the spray area. Lateral resolution is limited by sensitivity of the IMS, and needs to be taken into account when assessing the data. In general, it can be enhanced by oversampling, but, more preferably, it should be measured against a reference material [23,24].

### 3.1. MALDI

MALDI is the most prevalent ionization technique, due to its versatility in detecting metabolites ranging from the realm of hundreds of Da to 100 kDa and above [25]. In MALDI, the tissue samples are completely coated with a laser-absorbing matrix, typically an organic compound of low molecular weight, which allows the extraction of the metabolites from the sample, also known as the analytes, upon matrix crystallization after solvent evaporation (Table 2). The sample is then hit with a sufficiently energized laser to desorb and ionize the aggregate of sample and matrix molecules [25]. In the crystallized form, analyte degradation is reduced and the analyte molecules of larger molecular weight are better able to resist fragmentation, due to the energy-absorbing properties of the matrix molecules [22]. After the metabolites are desorbed and ionized into a gaseous form, the analyte ions are guided by ion lenses to be detected by a mass analyzer, generating a spectrum of mass to charge (*m*/*z*) ratios for all detected analytes [25] (Figure 1). MALDI is one of the most popular ionization techniques for IMS, because of the combination of its high sensitivity, capability of detecting a large range of analytes, tolerance for sample contaminants, and simplification of analysis due to the production of only singly-protonated or singly-deprotonated ions in positive and negative mode, respectively, and protection from fragmentation [21,22,25]. In addition to a traditional vacuum MALDI, atmospheric pressure (AP) MALDI can be used to analyze volatile compounds, as samples are not exposed to a high vacuum environment prior to analysis [26]. AP-MALDI has also been used more frequently over the past few years, particularly for imaging lipids [27], sugars, and peptides [28]. As it has a more focused beam, it produces images with higher resolution compared to vacuum MALDI, which is very important for single cell IMS [29]. 

### 3.2. DESI

With desorption electrospray ionization (DESI), ionization of the sample is possible under atmospheric pressure, unlike the vacuum-induced high pressure environments in which MALDI and SIMS must be carried out [22]. DESI swaps out the laser- and energy-absorbing matrix used in MALDI for electrically charged solvent spray and solvent ion droplets, which are then directed onto the sample surface [45]. The impact of these projectile-like droplet particles provides the energy to eject the analyte molecules into a gaseous state via electrostatic and gaseous forces [46]. After desorption, the singly- or multiply-charged analyte ions are collected through the atmospheric inlet line of a mass spectrometer for subsequent detection [22,46]. By avoiding the rough, high-pressure environment prevalent in most ionization techniques, DESI offers the benefit of conducting sample ionization in a relatively gentle environment [47]. This leads to a higher chance of detecting intact molecular ions rather than fragment ions, resulting in a less complex mass spectrum. Secondly, by not requiring sample preparation procedures such as matrix application, analyses involving DESI ionization avoid problems associatedwith non-homogenous matrix coating or the selection of the optimal matrix and solvent to analyte compatibility [48]. Ionization via DESI can also be preferred for certain classes of molecules; for example, lipids and proteins are generally ionized more effectively by DESI and MALDI, respectively [45,49,50]. The major drawback of DESI, however, is low spatial resolution [50], which limits its widespread use.

### 3.3. SIMS

Secondary ion mass spectrometry (SIMS) employs the use of a high energy primary ion beam to cause the desorption and ionization of analytes from the sample tissue in the form of secondary ions [22]. One of the most common primary ion sources utilizes high-energy gallium and indium ions [51]. The surplus of energy from these primary ion beams causes high rates of fragmentation to the analytes, producing secondary ions that are accelerated through a mass analyzer until they hit the detector [22]. The main advantage of SIMS is the extremely high spatial resolution, as it can reach a resolving power on the scale of the micron [50]. However, damage to the sample upon impact of the primary ion beam is unavoidable, and, after a certain amount of ionization damage, there is no longer any detectable signal from the emission of the secondary ion [51]. Furthermore, due to its dependence on secondary ion emission, SIMS is ineffective at detecting certain metabolites, specifically hydrophilic metabolites that are present only in low concentrations [52]. Extensive molecular fragmentation often complicates post-ionization analysis, and analytes of high molecular weight are eradicated [22]. However, the development of new primary ion beams with relatively lower energies has been successful in preserving the intactness of higher molecular weight analytes, thereby improving the efficacy of SIMS for a broader range of metabolites of interest [53].

## 4. IMS Analysis

After the ionization and subsequent detection of the various analytes, an *m*/*z* spectrum corresponding to the entire range of detected metabolites is generated. Software such as *HDImaging* (Waters Corp., Milford, MA, USA) can be used to view and sort these metabolites by selecting *m*/*z* values to visualize the spatial distribution of that particular metabolite, overlaid on the anatomical image of the sample slice. Normalization by total ion current (TIC) is generally done to account for the possible variance in the sample. If searching for a particular metabolite, online databases (such as http://www.hmdb.ca/) can be used to get accurate *m*/*z* value for the metabolite of interest, and then to search for that value in the IMS spectrum, to obtain its spatial distribution. Note that the metabolite of interest is not always present, but this does not necessarily mean that it is not present in the sample. Namely, the concentration might be too low, or a non-ideal sample preparation resulted in destruction of the metabolite. In addition to visualization, quantification of metabolite levels can be performed by selecting the region of interest and extracting the raw data, which can then be further processed with available software, such as *Progenesis QI* (Waters Corp, Milford, MA, USA). In this way, metabolite distributions can be compared between regions of interest and non-interest to illuminate the contribution of given metabolites to the physiological differences of given regions.

## 5. Neurodegenerative Disorders

Over the past few decades, neurodegenerative diseases have taken the front stage as one of the largest public health concerns. No quantitative biomarkers exist to enable their early diagnosis and commencement of therapy. Furthermore, therapy is non-specific and neurodegeneration progresses, eventually leading to death. The 4.7 million Americans estimated to have Alzheimer’s disease (AD) in 2010 has been projected to increase to 13.8 million by 2050 [54] (Figure 2). Worldwide, about 35.6 million people are estimated to be living with dementia, anticipated to increase to much higher rate as low income countries become more developed [55]. In parallel, Parkinson’s disease (PD), the second most common neurodegenerative disease, has shown similar increases in prevalence. The 680,000 Americans aged 45 and older who suffered from PD in 2010 is expected to increase to more than 1 million Americans by 2030 [56]. Thus, it is imperative to continue research toward preventing the rising tide of neurodegenerative diseases. Metabolic dysfunction has been reported in most of these diseases as we describe in more detail below, focusing on recent data from IMS studies. 

### 5.1. Alzheimer’s Disease (AD)

AD is a progressive neurodegenerative disease that causes gradual loss of memory and other cognitive abilities, as well as emotional and behavioral deficits [17,57]. In addition to the progressive nature of AD, advancing at different rate for each affected individual, its multitude of symptoms can change over time as different brain regions undergo pathology [58]. Common symptoms of AD include loss of explicit memory abilities, difficulties with problem-solving and planning, complications with temporal processing, struggling with writing, alterations in mood and personality, and social withdrawal as difficulties arise with maintaining speech flow during conversations. In the most severe stages of AD, affected individuals become non-ambulatory, and it is their bed-ridden nature that makes them especially susceptible to blood-clots and infectious disease. Eventually, those with AD will succumb to organ failure or aspiration pneumonia, due to chronic infection or accidental ingestion of food into the lungs, respectively. 

The complex pathogenesis of AD involves aberrant processing of amyloid-β proteins and the hyperphosphorylation of tau proteins, which aggregate to form amyloid-β plaques, amyloid angiopathy, and tau protein neurofibrillary tangles [57,59]. These aggregations lead to neural degeneration in the hippocampus and entorhinal cortex, the centers of learning and memory in the brain [59,60]. Additionally, the loss of neurons and synaptic activity, oxidative stress, and changes in the activity of reactive glial cells contribute to the gradual decline of cognitive functioning [17,57,59]. Besides protein aggregation, irregular lipid metabolism in the brain has also been implicated in the pathogenesis of AD [61,62]. Lipids are involved in cell signaling pathways and function as the building blocks of cellular membranes; thus, investigations of differential lipid activity in the AD has become a topic of great interest. Furthermore, there are connections to be made between faulty protein and lipid metabolism, as amyloid-β proteins prompt neuronal damage by moderating phospholipase activity [62]. Further, abnormal lipid accumulation may also play a critical role in neurodegeneration [16]. Recent evidence has also implicated impaired adult neurogenesis in the development of the disease, as mouse models of AD established decreased rates of adult neurogenesis [60,63,64,65], recently confirmed in humans as well [66]. Consequently, it is hypothesized that AD pathogenesis may be related to the effects of aggregates on both mature neuronal death and neural progenitor cell (NPC) activity within the neurogenic niche of the dentate gyrus [67]. Indeed, several computational models have been used to predict the effects of apoptosis on neurogenic potential in both the young and aged brain [65,66,68]. With the recent seminal study on postmortem brains from AD patients [66], it is clear that more research should be done to examine the role of neurogenesis as a possible therapeutic target for AD. 

The first studies exploring metabolic alterations in AD used conventional mass spectrometry (MS)-based metabolomics [17]. For example, gas chromatography–mass spectrometry (GC-MS) and ultra-high performance liquid chromatography–mass spectrometry (HLPC-MS) were used to examine transgenic AD mouse models. These studies found that the AD brain exhibited significant metabolomic differences compared to the wild-type mouse control brain [69]. Although the differences existed primarily in the hippocampus and cortex, other regions not traditionally associated with the disease, such as the striatum, cerebellum, and olfactory bulb, were affected as well. These results pointed to AD pathology stemming from myriad dysregulations across several different metabolic pathways [69]. The contributing metabolites to the abnormal mice neurochemical profile were identified as phospholipids, fatty acids, purine and pyrimidine metabolites, sterols, and others [70]. Similar results were obtained when conducting MS analysis of the human AD brain [71]. Through the investigation of seven neural regions, partitioned into the categories of most damaged, moderately damaged, and lightly damaged by AD pathology, the levels of 55 total metabolites were confirmed to be altered in at least one of those regions [71]. The wide range of regions in which these metabolic abnormalities were seen supports the theory of whole-brain degeneration in AD [71], which could also be contributed to by the regional differences in metabolism, as reported in the mouse brain [14]. 

To correlate the spatial distribution of various biomolecules with the amyloid aggregates within the AD brain, MALDI-IMS found sphingolipid, phospholipid, and lysophospholipid changes associated with individual plaques within tissue samples [72]. These correlative data indicate a possible loop between amyloid aggregates and changes in metabolites that reflect inflammation, oxidative stress, demyelination, and cell death [72] (Table 3). In addition, specific investigations into lipidomic dysfunctions within the human AD brains incorporated the use of MALDI-IMS in both positive and negative ion modes, depending on the relative ease with which lipids were protonated or deprotonated, to identify alterations in different classes of lipids within the hippocampus [73]. In both the control and AD brains, positively and negatively charged ions were readily detected. Although the distribution of the lipid species was consistent across all samples, their relative abundance differed, predominantly in the CA1 region and dentate gyrus of the hippocampus [73].

Besides the neurogenic areas, MALDI-IMS was also conducted on the frontal cortices of postmortem human brain of AD patients, which were further subcategorized into increasing disease severity as AD I–VI, using Braak’s histochemical criteria. These studies showed that sulfatide concentration begins to decrease in the early stages of AD in the white and gray matter of the frontal cortex [63]. As these differences were observed only in AD, and not in other neurodegenerative disorders, sulfatide is suspected to play a role in AD pathology, and may be used as a biomarker for AD [61]. Another class of lipids believed to be important for AD pathology are sphingolipids, which include ceramides, sulfatides, and gangliosides [72]. Since their localization is important to validation of this claim, MALDI-IMS was once again utilized to elucidate the sphingolipid spatial profile and its association with the amyloid plaques [72]. IMS revealed that sphingolipids selectively localized to beta-amyloid aggregates within the cortex and hippocampus. Specifically, gangliosides and ceramides were highly localized to beta-amyloid positive plaques, suggesting that differences in sphingolipid concentration may be important to AD pathogenesis [72]. More recently, MALDI-IMS analysis of the lipid signatures in transgenic AD mouse models showed early shifts in lipid homeostasis that commenced early in AD pathology [74]. Namely, the white matter of AD mouse brains contained significantly lower levels of complex gangliosides, such as the GD1 d18:1 species, and higher levels of simple gangliosides in the prodromal phases of AD [74]. 

Another technique, laser ablation ICP-MS, has also been used to examine the role of iron in AD [75,76]. Unlike MALDI-IMS, ICP-MS vaporizes the sample spot, and the resulting vapors are guided to a sample inlet for ionization by an ICP torch. As such, the ICP-MS breaks all sample molecules into their elemental form. Consequently, this method is excellent for detecting metal ions within samples. The inherent disadvantage of the ICP-MS lies in the analysis of biological components such as lipids, as the technique needs a metal-attached label to its intended target before analysis can be performed. Since iron buildup is implicated in the development of AD, due to iron’s capacity to increase oxidative stress, ICP-MS studies were conducted on AD brains, revealing higher levels of iron in the gray matter in the AD frontal cortex [76,77].Recently, higher levels of iron were found within the CA1 region of the hippocampus [75], but this was not accompanied by a corresponding increase in the ferroportin transport protein [75]. Regardless, IMS has provided new data with respect to iron upregulation in AD, further supporting the oxidative stress hypothesis of AD [75,77].

### 5.2. Parkinson’s Disease (PD)

Parkinson’s disease (PD) is a progressive neurodegenerative disease characterized by the death of dopaminergic neurons located within the substantia nigra [78]. These neurons form the nigrostriatal dopaminergic pathway; their death results in a dramatic decrease in levels of dopamine in the striatum that results in clinical presentation such as a resting tremor of varying intensity, unstable posture, muscle rigidity, freezing (inability to initiate voluntary movement), and voluntary movement that is characteristically slow and decreased in intensity. PD also affects cognitive abilities and leads to dementia, as neurodegeneration eventually starts to affect the nearby hippocampal and cortical regions [78].

MALDI imaging studies have often been conducted on PD animal models generated through the injection of either 1-methyl-4-phenyl-1,2,3,6-tetra-hydropyridine (MPTP) or 6-hydroxydopamine (6-OHDA) into one hemisphere of the rodent brain, allowing the contralateral side to serve as an internal control [21] (Table 3). MALDI imaging of 6-OHDA mice revealed the unusual presence of collapsin response mediator protein 2 (CRMP-2) in the corpus callosum—this protein is usually found only in the dendrites of hippocampal and cortical CA1 pyramidal cells or Purkinje cerebellar cells [79]. The hyperphosphorylation of certain amino acid residues on CRMP-2 has been associated with AD neural degeneration; this study pointed to the possibility of PD dyshomeostasis involving similar mechanisms, and the potential use of CRMP-2 as biomarkers for PD as well as AD [79,80,81]. Another MALDI-IMS analysis of MPTP mice localized PEP-19, a calmodulin-binding protein, in the striatum of the control brain, but found a significant reduction in the PD brain [82]. Further, MALDI-IMS was used to examine L-DOPA-induced dyskinesia, a common side effect of PD medications [83,84,85]. L-DOPA, the precursor of dopamine, is commonly used in the treatment of PD, due to its efficacy in reducing many of the symptoms associated with the neurodegenerative disorder [84]. However, as the medication is taken for years, some patients develop dyskinesia [84]. Interestingly, MALDI-IMS indicated not only a positive correlation between the severity of L-DOPA-induced dyskinesia and the nigral levels of dynorphin B and alpha-neoendorphin, but also that the most significant differences were localized to the lateral substantia nigra [83], suggesting possible mechanisms that might be amenable to targeted treatment. 

### 5.3. Huntington’s Disease (HD)

Huntington’s disease (HD) is a progressive, autosomal-dominant neurodegenerative disease caused by a trinucleotide (CAG) repeat expansion of the gene encoding the huntingtin protein. The resultant mutant protein is believed to be neurotoxic, leading to the destruction of medium spiny neurons within the striatum and resulting in clinical symptoms such as dyskinesia, neuropsychiatric symptoms, and cognitive impairment [90,91]. However, recent evidence also points to a wider extent of neurodegeneration that extends into the cortical areas of the brain, which may explain the diversity of HD clinical presentations [91]. Later stages of HD are often characterized by severe motor impairment and dementia, with judgement, reasoning, and comprehension abilities also experiencing a great loss [91]. As the misfolded proteins accumulate, cellular degradation mechanisms are not able to keep up and eliminate the toxic huntingtin aggregates. However, the exact process by which polyglutamine aggregation causes the selective destruction of neurons remains to be determined [92].

In the past, GC-MS of human HD brains pinpointed urea upregulation as a possible causal factor of HD neurodegeneration. Further, GC-MS studies pointed to a widespread metabolic dysregulation in brain regions that extends outside of canonically damaged regions of the HD brain [93,94]. This was confirmed with direct injection liquid chromatography mass spectrometry (DI/LC-MS/MS) studies on postmortem brains of HD patients, which revealed metabolic differences in both the frontal lobe and striatum [95]. Particularly, the HD brain contained significantly lower endogenous levels of acylcarnitine, the neuroprotective compounds vital for proper energy metabolism, and phospholipids, which are important in both cellular signaling and integrity [95].

Contrary to the findings observed in AD and PD, adult neurogenesis in the HD brain surprisingly exhibits an increase, reported within the subventricular zone, one of two regions where adult neurogenesis occurs [92]. In addition, MALDI-IMS studies of the lipidome indicated significant differences between control and HD brains in the myelin and ependymal layer [86] (Table 3). The HD myelin layer had decreased sulfatides and triglycerides, as well as enrichment of sphingomyelin and ceramide-1-phosphates. The alterations in the HD ependymal layer were mainly attributed to a drastic drop in the concentration of phosphatidylinositols [86]. 

Finally, in a study evaluating the efficacy of a 23 amino acid peptide sequence known as P42 in exerting a neuroprotective effect on HD mouse models, MALDI-IMS was employed to examine the pharmakokinetics of P42 delivery into the in vivo model, which included the spatial distribution of P42 and its degraded products, the extent to which it was able to reach the target site, and rates of diffusion between various neuronal compartments [87]. With the aid of the imaging technology, the investigators obtained valuable information on the efficacy of drug delivery and targeting of the neurons within the striatum that correlated with the subsequent improvement in performance on behavioral tests and decreases in protein aggregation, leading to an overall improvement of HD symptoms in this model [87].

### 5.4. Multiple Sclerosis (MS)

Multiple sclerosis (MS) is a complex, chronic, immune-mediated demyelinating disease affecting the central nervous system in which primary inflammation leads to secondary neurodegeneration. It is the most commonly acquired complex degenerative brain disease of young adults, and is among the most frequent causes of disability in early to middle adulthood [96,97]. The disease course varies, and presents with unpredictable symptoms and levels of recovery. Most commonly, patients are diagnosed with Relapsing-Remitting MS (RRMS), which eventually progresses to the Secondary Progressive (SPMS) form and patients rapidly decline. In the pathogenesis of MS, resident microglia and astrocytes, together with infiltrating macrophages and T-and B-lymphocytes, become activated and produce large amounts of inflammatory cytokines, prostaglandins, and other toxins, that lead to demyelination and ultimately to axonal degeneration. The cumulative effects of these mechanisms increase neurodegeneration within the MS brain and eventually lead to the worsening of clinical symptoms, including cognitive deficits, depression, upper motor neuron signs, tremors, fatigue, weakness, and pain [96,97]. The management of MS is plagued by the variability of the clinical course and severity. This poses great difficulty in providing any given individual with an accurate prognosis and customized treatment plan. Unfortunately, there are no biomarkers able to indicate when the next relapse would occur, whether the given therapy would work, and whether the disease process will last several years or several decades. This uncertain aspect of the disease adds to the health care costs and the emotional distress associated with the disorder.

In the past, human MS brain tissue analyzed by electrospray ionization tandem mass spectrometry showed that brains with active MS demonstrated increases in phospholipid levels and decreases in sphingolipid levels in the normal appearing white matter and gray matter. These changes in lipid signatures could result from metabolic dysregulation that causes sphingolipids to be shuttled into synthesizing phospholipids [98]. Additionally, GC-MS showed significant alterations in the abundance of 44 different metabolites in the MS brain, which were traced back to metabolic intermediates integral to biochemical pathways such as bile acid biosynthesis, taurine metabolism, tryptophan and histidine metabolism, linoleic acid, and D-arginine metabolism pathways [99].

To localize metabolic differences reported by GC-MS and other methods, MALDI-IMS was used for the analysis of recurrent inflammatory lesions in post-mortem MS brains to elucidate the spatial distribution of proteins and peptides [88] (Table 3). Specifically, the objective was to characterize the proteins that were only expressed in the intact white and gray matter, as well as the ones preferentially localized to inflammatory lesions. Analysis revealed that thymosin beta-4, a protein involved in cellular migration, proliferation, and differentiation, was localized to active lesions that were chronically demyelinated and lesions that were only partially remyelinated, suggesting a neurorestorative function for the protein to facilitate remyelination through a downregulation of inflammation and upregulation of oligodendrocyte activity in damaged areas. This spatial confirmation of endogenous relevance to the MS disease pathology allowed researchers to conclude that thymosin beta-4 played a neuroprotective and neurorestorative role in the demyelinated CNS [88]. 

### 5.5. Amyotrophic Lateral Sclerosis (ALS)

Amyotrophic lateral sclerosis (ALS) is a neurodegenerative disease that involves progressive deterioration of motor neurons located in the motor cortex, brainstem, and spinal cord [100]. As both upper and lower motor neurons begin to degenerate, the muscles grow weak from disuse, exhibit fasciculations, and eventually begin to atrophy. Although 10% of ALS diagnosis have been linked to genetic factors, such as a mutation in the copper–zinc superoxide dimutase 1 (SOD1) gene, the disease is generally regarded to be idiopathic, because the overarching biological pathways that lead to ALS are still unknown [101]. Current proposed disease mechanisms for ALS include intricate relationships between cell-damaging gain of function by SOD1, inflammation from microglial activity, intracellular aggregates, defective mitochondrial function, and glutamate-induced excitotoxicity resulting in neurodegeneration and free radical production [102]. However, since no absolute clinical or molecular biomarkers for the disease exist, especially for idiopathic patients that do not carry mutations in SOD1, combined with the similarity of its initial symptoms with other neurological disorders, the accurate diagnosis and prognosis of ALS is difficult [89]. One application of MALDI-IMS towards investigating this neurodegenerative disease focused on post-mortem human spinal cords from both ALS and control patients [89] (Table 3). Control spinal cords had normal metabolite distributions, some of which included histone and thymosin beta-4 proteins. Contrarily, ALS spinal cords contained significantly lower gray matter concentrations of a truncated form of ubiquitin (Ubc-174) in which both C-terminal glycine residues had been removed. The specific localization of the proteins showed that alterations in protease activity validated the hypothesis that proteome dysfunction plays a significant role in ALS pathology [89].

## 6. Conclusions

With neurodegenerative diseases on the rise all around the world, a lack of understanding of their causative factors continues to contribute to their personal, societal, and financial burden as the aging population grows ever larger. To date, the inability to find biomarkers and cures for the wide spectrum of neurodegenerative disorders can be partially attributed to the lack of analytical technologies and methods that incorporate both the requisite specificity and sensitivity to study the human brain as the primary site of these complex and devastating diseases. Recent advancements in metabolomics have seen the rapid surge of the use of MS to facilitate studies of both the proteome and the lipidome, with the hope that accurate diagnostic and prognostic biomarkers can be identified for early detection and initiation of preventative measures. Furthermore, utilization of IMS has provided spatial localization for those metabolites detected by IMS, solidifying their role in many cases. Thus, IMS, along with other technologies available today, is paving the way for elucidation of metabolic dysregulation and neurodegenerative dyshomeostasis, towardsthe discovery of new targets for precision therapy of these disorders.

## Figures and Tables

**Figure 1 metabolites-09-00135-f001:**
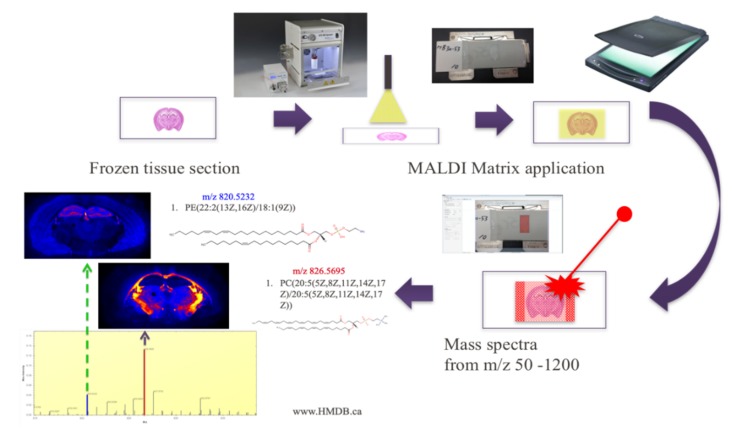
Ion mass spectrometry (IMS) experimental workflow with MALDI. The frozen tissue section must be coated with a matrix and ionized by the laser before metabolite localization within the section can be mapped.

**Figure 2 metabolites-09-00135-f002:**
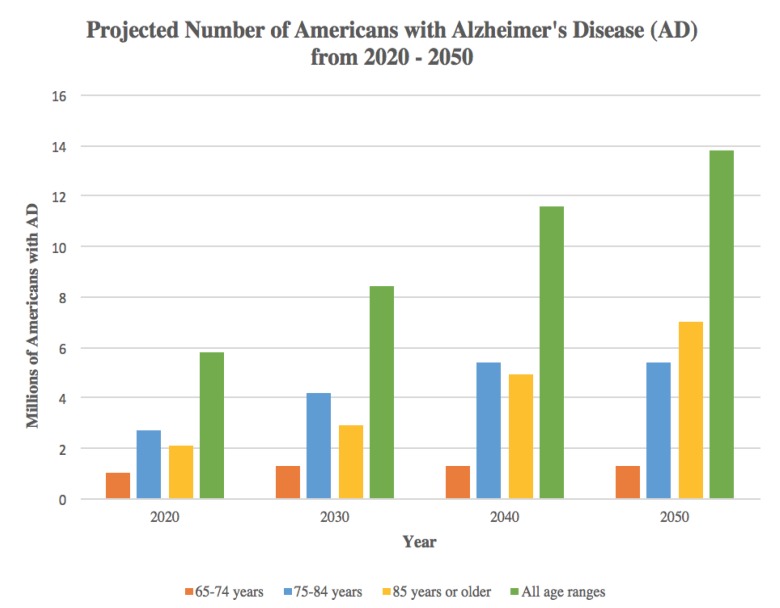
Projected number of Americans with Alzheimer’s disease (AD) from 2020–2050. Source: Created from data in Herbert LE, Weuve J, Scherr PA, Evans DA. Alzheimer disease in the United States (2010–2050) estimated using the 2010 Census, *Neurology* 2013; 80(19):1778–1783 [54].

**Table 1 metabolites-09-00135-t001:** Three main ionization techniques for mass spectrometry: matrix-assisted laser desorption/ionization (MALDI), desorption electrospray ionization (DESI), and secondary ion mass spectrometry (SIMS).

Platform	Mechanism	Advantages	Disadvantages
Matrix-assisted laser desorption/ionization (MALDI)	UVlaser used for the desorption and ionization of analytes after application of an UV-absorbing matrix into a gaseous state	High sensitivity Tolerates sample contaminants and impurities Detects a large range of *m*/*z* values Simple post-ionization analysis due to generating only singly-protonated or singly deprotonated ions (depending on whether positive or negative mode is used) Low fragmentation rate allows analysis of metabolites of large relative molecular weight	Matrix application must be uniform to minimize artifacts during post-ionization analysis and imaging Matrix application needs to be tailored for each tissue type to ensure optimal coverage and thickness Overcoating of matrix reduces detected signal. Internal standard is needed for calibration purposes
Desorption electrospray ionization (DESI)	Electrically-charged solvent drops sprayed onto sample surface to eject analyte molecules into a gaseous state	Extremely high spatial resolution (resolving power capable of reaching the micron scale) Focusing capabilities of primary ion beams are superior to those of lasers	Lower spatial resolution compared to other methods of ionization
Secondary ion mass spectrometry (SIMS)	High-energy primary ion beam (i.e., gallium and indium ions) facilitates the desorption and ionization of analytes in the form of secondary ions in the gaseous state	Extremely high spatial resolution (resolving power capable of reaching the micron scale) Focusing capabilities of primary ion beams are superior to those of lasers	Significant damage done to sample upon primary ion beam impact Ineffective at detecting certain types of metabolites High rates of molecular fragmentation complicates post-ionization imaging and analysis Inability to analyze metabolites of higher relative molecular weight (e.g., >1000 *m*/*z*)

**Table 2 metabolites-09-00135-t002:** The different types of matrices used in MALDI and their application.

Matrix	Application	References
2-amino-5-nitropyridine (ANP)	Oligonucleotides < 20 Bases, MALDI (−)	[30,31]
80% anthranilic acid + 20% nicotinic acid (80/20 AA/NA)	Oligonucleotides < 20 Bases, MALDI (+ & −)	[32,33]
6-aza-2-thiothymine (6-ATT)	Oligonucleotides and carbohydrates MALDI (−)	[34,35]
3-hydroxypicolinic acid (3-HPA)	Oligonucleotides 1 kDa−30 kDa	[36,37]
α cyano 4 hydroxycinnamic acid (CHCA)	small molecules, peptides/proteins < 6 kDa	[38,39,40]
2,5 dihydroxybenzoic acid (2,5 DHB)	small molecules, peptides/proteins < 6 kDa, polymers, carbohydrates	[39,41,42,43]
9-aminoacridine (9-AA)	small molecules, lipids, MALDI (−)	[44]

**Table 3 metabolites-09-00135-t003:** Application of IMS to various neurodegenerative diseases.IMS has been used to study the metabolomics of Alzheimer’s disease, Parkinson’s disease, Huntington’s disease, multiple sclerosis, and amyotrophic lateral sclerosis in both humans and mouse models. Only MALDI-IMS has been used.

Disease	Organism	Findings	Ref.
Alzheimer’s Disease (AD)	Humans	Relative abundance of lipid species differed between AD and control brains, predominantly in the CA1 and dentate gyrus regions of the hippocampus.	[73]
-	Humans	Sulfatide concentrations start to decrease during the early stages of AD (determined by Braak’s histochemical criteria) in the white and gray matter of the frontal cortex.	[61]
-	Humans	Sphingolipids (e.g., ceramides, sulfatides, and gangliosides) show selective localization to the β-amyloid aggregates within the cortex and hippocampus, with specific localization of gangliosides and ceramides to β-amyloid positive plaques.	[72]
-	Humans	The CA1 region of the hippocampus in the AD brain contains higher levels of iron compared to the control brain, but there is no significant difference in the levels of ferroportin transport protein between AD and control brains.	[75]
-	Mice	Lipid signatures of the AD mouse brain exhibit early shifts in lipid homeostasis: its white matter is composed of higher levels of simple gangliosides and lower levels of complex gangliosides, such as the GD1 d18:1 species.	[74]
Parkinson’s Disease (PD)	Mice	Collapsin response mediator protein 2 (CRMP-2) detected in the PD brain though usually only found in hippocampal and cerebellar cells.	[79]
-	Mice	Calmodulin-binding protein (PEP-19), normally localized to the striatum, was significantly downregulated in the stratium of the PD brain.	[82]
-	Mice	Distinct differences in both the levels and localization of various neurotransmitters and amino acid between PD and control brains established.	[85]
-	Mice	Dynorphin B and alpha-neoendrophin nigral levels are positively correlated with the severity of L-DOPA-induced-dyskinesia.	[83]
Huntington’s Disease (HD)	Mice	Within the HD myelin layer, sulfatide and triglyceride levels are decreased and sphingomyelin and ceramide-1-phosphate levels are increased in the lamina; within the HD ependymal layer, phosphatidylinositols levels are decreased.	[86]
-	Mice	Efficacy of P42, a 23 amino acid peptide sequence, as a novel therapy for HD was analyzed with IMS to confirm drug delivery, investigate pharmokinetic properties, and observe post-delivery molecular change.	[87]
Multiple Sclerosis (MS)	Humans	Thymosin beta-4 protein localized to active MS lesions that were either chronically demyelinated or only partially remyelinated.	[88]
Amyotrophic Lateral Sclerosis (ALS)	Humans	Truncated ubiquition form (Ubc-174) levels decreased significantly in ALS spinal cords compared to control, which paralleled normal histological distributes of metabolites in the gray matter.	[89]

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
