# Peer review of "Imaging Mass Spectrometry: A New Tool to Assess Molecular Underpinnings of Neurodegeneration"

_metabolites, 2019, doi:10.3390/metabo9070135_

Round 1

Reviewer 1 Report

The review by Chen et al. is well articulated and in my opinion could be accepted for publication in Metabolites. However, I suggest some revisions before publications:

1)      The sentence “The main advantages of IMS that have allowed it to gain it recent prominence center around its ability to detect…” is wrong;

2)      In the section discussing MALDI, I would add a part describing all the possible matrices used in MALDI according to the analyte of interest (maybe a table could help as well); moreover, more studies where such approach has been used should be cited, to give to the reader some examples;

3)      What is the lateral resolution in MALDI and DESI? I think lateral resolution needs to be better discussed and relevant works need to be cited;

4)      More citations regarding DESI are advised (there are many more than the once cited! See, for example a very recent one: Analytical Chemistry (2019), 91(4), 2719-2726);

5)      What about Atmospheric pressure (AP) MALDI?

6)      The sentence “In low or middle-income countries, care of individuals affected by dementia are usually done by family members and are this informal care is mostly unpaid” is wrong;

7)      It seems that the authors have decided to discuss some of the main actors involved in AD, PD, etc. What about metal ions? What about ICP-MS (NeuroImage (2016), 137, 124-131.)?

Author Response

We highly appreciate the detailed review and constructive comments from the Reviewer 1. We have corrected the manuscript based on the comments and added more information as the reviewer asked for. We almost doubled the number of references, which contributed to the extension of time for our resubmission. As we have substantially corrected the text, we did not highlight the changed parts for clarity of reading.

1)      The sentence “The main advantages of IMS that have allowed it to gain it recent prominence center around its ability to detect…” is wrong;

We eliminated this statement.

2)      In the section discussing MALDI, I would add a part describing all the possible matrices used in MALDI according to the analyte of interest (maybe a table could help as well); moreover, more studies where such approach has been used should be cited, to give to the reader some examples;

We have now included the table with relevant references, please see Table 2.

3)      What is the lateral resolution in MALDI and DESI? I think lateral resolution needs to be better discussed and relevant works need to be cited;

We have commented on lateral resolution in Section 3. Ionization methods, please see page 5, second paragraph.

4)      More citations regarding DESI are advised (there are many more than the once cited! See, for example a very recent one: Analytical Chemistry (2019), 91(4), 2719-2726);

We have expanded the description and references on DESI, including the most recent paper. Please see page 6, Section 3.2. DESI.

5)      What about Atmospheric pressure (AP) MALDI?

We have added AP-MALDI description, please see page 6, first paragraph.

6)      The sentence “In low or middle-income countries, care of individuals affected by dementia are usually done by family members and are this informal care is mostly unpaid” is wrong;

We eliminated this statement

7)      It seems that the authors have decided to discuss some of the main actors involved in AD, PD, etc. What about metal ions? What about ICP-MS (NeuroImage (2016), 137, 124-131.)?

We have included a section on the ICP-MS in the discussion of AD, please see Section 5.1., page 10, second paragraph.

Reviewer 2 Report

This is a very interesting review on the application of imaging mass spectrometry (IMS) based metabolomics for the investigation of molecular mechanisms associated to neurodegerative disorders. This is a hot topic nowadays, whose potential is still largely unexplored. The manuscript is clear and well presented, and authors include a lot of relevant literature in this field. I only suggest some minor modifications before publication.

- Authors introduced a long description about Alzheimer’s disease in section 5, which should be moved to section 5.1

- Authors should mention than conventional MS-based metabolomics has also been employed to elucidate metabolomics alterations associated to neurodegenerative diseases in various dissected brain regions. These results must be compared with findings from IMS. Some examples: Alzheimer’s disease (J Pharm Biomed Anal 2015,102:425-435; Biochim Biophys Acta 2016,1862:1084-1092; Biochim Biophys Acta 2014,1842:2395-2402), Huntington’s disease (Biochim Biophys Acta 2018,1864:2430-2437; Biochim Biophys Acta 2016,1862:1650-1662; Biochem Biophys Res Commun 2015,468:161-166)

Author Response

We would like to thank Reviewer 2 for the appreciation of the work we have done and for the suggestion to include some conventional MS-based metabolomics data on neurodegenerative diseases. We have published such review (Botas et al., 2015; PMID: 26358890) and cited it here. In addition, we reviewed more literature published since then. Unfortunately, we could compare only a handeful of studies because most conventional MS was done on biofluid samples and not on postmortem tissue; still, we mention the work as it is highly converging.

Authors introduced a long description about Alzheimer’s disease in section 5, which should be moved to section 5.1

We agree and moved the text to Section 5.1. The introductory part to neurodegenerative diseases is now more straightforward.

- Authors should mention than conventional MS-based metabolomics has also been employed to elucidate metabolomics alterations associated to neurodegenerative diseases in various dissected brain regions. These results must be compared with findings from IMS.

We have referred to conventional MS-based metabolomics in each discussed disease - please see relevant paragraphs in those sections.